# Using unsupervised machine learning to quantify physical activity from accelerometry in a diverse and rapidly changing population

**Christopher B. Thornton**[1]*, **Niina Kolehmainen**[1,2], **Kianoush Nazarpour**[3]

**1** Population Health Sciences Institute, Faculty of Medical Sciences, Newcastle University, United Kingdom, **2** Great North Children's Hospital, Newcastle upon Tyne NHS Hospitals Trust, Unite Kingdom, **3** Institute for Adaptive and Neural Computation, School of Informatics, The University of Edinburgh, United Kingdom

\* chris.thornton@newcastle.ac.uk

**Data Availability Statement:** All data underlying the results presented in this paper is available on the data.ncl repository. Physical activity data and the fitted HSMM model is available at doi.org/10.

## Abstract

Accelerometers are widely used to measure physical activity behaviour, including in children. The traditional method for processing acceleration data uses cut points to define physical activity intensity, relying on calibration studies that relate the magnitude of acceleration to energy expenditure. However, these relationships do not generalise across diverse populations and hence they must be parametrised for each subpopulation (e.g., age groups) which is costly and makes studies across diverse populations and over time difficult. A data-driven approach that allows physical activity intensity states to emerge from the data, without relying on parameters derived from external populations, offers a new perspective on this problem and potentially improved results. We applied an unsupervised machine learning approach, namely a hidden semi-Markov model, to segment and cluster the raw accelerometer data recorded (using a waist-worn ActiGraph GT3X+) from 279 children (9–38 months old) with a diverse range of developmental abilities (measured using the Paediatric Evaluation of Disability Inventory–Computer Adaptive Testing measure). We benchmarked this analysis with the cut points approach, calculated using thresholds from the literature which had been validated using the same device and for a population which most closely matched ours. Time spent active as measured by this unsupervised approach correlated more strongly with PEDI-CAT measures of the child's mobility ($R^2$: 0.51 vs 0.39), social-cognitive capacity ($R^2$: 0.32 vs 0.20), responsibility ($R^2$: 0.21 vs 0.13), daily activity ($R^2$: 0.35 vs 0.24), and age ($R^2$: 0.15 vs 0.1) than that measured using the cut points approach. Unsupervised machine learning offers the potential to provide a more sensitive, appropriate, and cost-effective approach to quantifying physical activity behaviour in diverse populations, compared to the current cut points approach. This, in turn, supports research that is more inclusive of diverse or rapidly changing populations.

25405/data.ncl.21196285. PEDI-CAT data is available at doi.org/10.25405/data.ncl.21120571.

**Funding:** NK and CT received funding from Health Education England (HEE) / National Institute for Health Research (NIHR) for this project (ICA-SCL-2015-01-003). KN was supported by the Engineering and Physical Sciences Research Council (EPSRC) under grant EP/R004242/2. The funders had no role in study design, data collection and analysis, decision to publish, or preparation of the manuscript.

**Competing interests:** The authors have declared that no competing interests exist.

## Author summary

Physical activity participation in young children has often been measured using parent reports. Accelerometry provides a more objective measurement but the traditional methods used to quantify this require calibration and struggle to generalise to diverse or rapidly changing populations such as young children. In recent years unsupervised machine learning methods have been shown to be able to segment and cluster accelerometry, allowing categories of activity intensity to emerge from a data-driven process. Here we show that an unsupervised machine learning technique (the hidden semi-Markov model) can be used to estimate categories of activity intensity in accelerometry data recorded from a diverse population of children age 9–36 months. We also show that this approach better captures the variance of movement abilities in the population than the traditional cut points approach. The hidden semi-Markov model approach provides a more effective approach for processing and analysing accelerometer data in rapidly changing and diverse populations such as young children, compared to the more traditional cut points approach. As it does not require calibration studies to incorporate new populations it has the potential to facilitate inclusion of unrepresented populations in research, as well as being less resource intensive.

## Introduction

Participation in physical activity is widely considered to be beneficial for all people. This includes young children for whom physical activity is known to promote development and positive health outcomes [1], while spending time sedentary has been shown to result in poor sleep [2]. Accurately measuring participation in physical activity by members of this age group, including those with non-typical developmental trajectories, is an important step in the development of interventions seeking to facilitate participation.

Accelerometers are increasingly used to measure physical activity. They record the acceleration of the body part to which they are attached, providing an objective record of how much movement has occurred. This raw acceleration recording can then be processed to extract the time spent in a range of physical activity intensity categories, such as "sedentary" (SED), "light physical activity" (LPA), and "moderate to vigorous physical activity" (MVPA). The traditional method used to process the raw acceleration trace into these categories, known as the cut points method, applies a threshold to the volume of acceleration recorded in each epoch [3]. This threshold is calibrated in a detailed lab-based study, where energy expenditure is measured at the same time as the acceleration, so that the cut point thresholds indicate the volume of acceleration at which the participant would be expected to expend a predefined level of energy. The energy levels of interest are usually calculated as a ratio of the energy expended while at rest–known as a Metabolic Equivalent (METs)–and for children are typically $<1.5$ METs for SED, 1.5–3 METs for LPA, and $>3$ METs for MVPA [4]. As the relationship between acceleration volume and energy expenditure depends upon the physical abilities, body size, and movement patterns of the child [5] calibration must be performed for each sub-population. This results in different cut points derived as children age and develop [6] and for children with different movement capabilities [7]. Using the cut-points approach is therefore challenging, if not impossible, when the population under study is diverse in their physical capabilities or when they are rapidly changing, such as in a longitudinal study of young children.

Recently, machine learning approaches have been increasingly applied to the analysis of accelerometer data. Some have sought to train supervised machine learning techniques to recognise activity types [8,9] while others have used unsupervised techniques to categorise activity based on the intensity and direction of movement [10–12]. This latter approach promises to segment and cluster the acceleration data according to its intensity, without parameters derived from external populations, and to offer a potentially more appropriate method for determining engagement in physical activity in longitudinal population studies of young children.

A HSMM approach has previously been used to segment and cluster accelerometer data recorded in children from the general population at age 14 [11]. In the present study we apply the hidden semi-Markov model (HSMM) to segment and cluster accelerometer data in 279 toddlers across abilities. We demonstrate that this method is a more appropriate approach in a rapidly developing and highly heterogeneous population. For comparison, we process the accelerometer data according to the traditional cut-points approach, using the best available parameters for this population. To evaluate which approach provides the most clinically relevant measure, we compare both approaches to how each child scores on the Paediatric Evaluation of Disability Inventory Computer Adaptive Test (PEDI-CAT), an established assessment of developmental capacities with strong psychometric properties across age groups and populations.

## Materials and methods

### Ethics statement

The data presented was collected as part of the ActiveCHILD project (NIHR ICA-SCL-2015-01-003) [13,14]. The study had NHS Research Ethics Committee and Health Research Authority (UK) approvals (NHS IRAS 218313, 17-NE-0051), and the design drew on the Nuffield Ethics guidance for health research involving children [15]. Formal written consent was obtained from the parents or guardians of all children who participated in this study.

### Data collection

Young children (n = 279, aged between 9 and 36 months) were recruited through the universal healthy child pathway (i.e. health visitors) and specialist children's health services (e.g. neonatology, community paediatrics, paediatric physiotherapy) in thirteen areas in England, UK. Different models of health visiting can be found across countries, under various labels [16]. Children recruited through specialist services (42%, n = 118/278) were purposefully oversampled to ensure coverage of children with a range of development trajectories. Table 1 shows the key characteristics of the sample; further details are available elsewhere [14].

To collect data, families were provided with pre-prepared accelerometer packs. The pack contained a pre-programmed accelerometer threaded on a flexible, waist-worn belt, and instructions [17] for the parent. Fig 1 illustrates how parents were instructed to fit the device. The device was set to record two days after the parent received the pack, and parents were encouraged to let their child play with the accelerometer on these pre-recording days in order to familiarise with the device. On the night before the first recording day, parents were asked to place the accelerometer beside the child's bed, to prompt them to put it on first thing in the morning. The parents were then asked to encourage the child to wear the accelerometer for seven days, except while bathing or showering, swimming, or in bed. At the end of the recoding period, parents were asked to post the accelerometer to the research team. In return of the accelerometer, parents were sent a feedback sheet of their child's activity and the child was sent a small toy as a reward. The ActiGraph GT3X+ was used, which has been previously

**Table 1. Shows the distribution of ages and recruitment pathways of the participants.**

|  | Total number of children | 279 |
|---|---|---|
| Recruitment (n = 278) | Health Visitor | 58% (160) |
|  | Other paediatric specialist | 42% (118) |
| Sex (% female) (n = 218) |  | 56% (123) |
| Age (months) (n = 277) | 5–10 | 5% (13) |
|  | 10–15 | 27% (76) |
|  | 15–20 | 18% (50) |
|  | 20–25 | 17% (47) |
|  | 25–30 | 22% (61) |
|  | 30–35 | 10% (28) |
|  | 35–40 | 1% (2) |
| Mobility as described by the clinician (n = 202) | Walks without support | 80% (162) |
|  | Uses walking aid | 10% (21) |
|  | Moves with support only (e.g. parent carrying, buggy) | 9% (19) |
| Cognitive development (Clinician's Assessment) (n = 203) | Unable to comment on the child's cognitive capacity | 19% (39) |
|  | No concerns about the child's cognitive development | 68% (139) |
|  | There are concerns about this child's cognitive development or learning (e.g. the child is below educational level, the child receives support for learning) | 9% (18) |
|  | The child has a global developmental delay established as part of a multidisciplinary or medical assessment | 3% (7) |

found to be acceptable and feasible to use in under5s [18] with physical limitations as well as typically developing children. The device was worn around the waist, set to record at 100 Hz, and set to capture record all movement that lasted at least one second.

The child's developmental abilities were measured using the Pediatric Evaluation of Disability Inventory—Computer Adaptive Test (PEDI-CAT). The PEDI-CAT is the current-generation version of a well-established, widely used measure Pediatric Evaluation of Disability Inventory (PEDI), originally published in 1992 and since revised as a computer adaptive test (CAT) [19]. The PEDI-CAT measures children's developmental abilities and disabilities, as manifested in daily life, across four domains (daily activities, mobility, social-cognitive, and responsibility). In our study, the PEDI-CAT was administered by the research team, with responses provided by one of the child's parents. The PEDI-CAT software uses Item Response Theory to estimate a child's abilities from a minimal number of the most relevant items within each domain, establishing the child's ability levels for each domain. Within the PEDI-CAT, the mobility domain assesses the child's movement behaviours such as sitting, crawling, and walking. The PEDI-CAT has been validated for use across diagnostic groups, ages, settings and countries. (19)

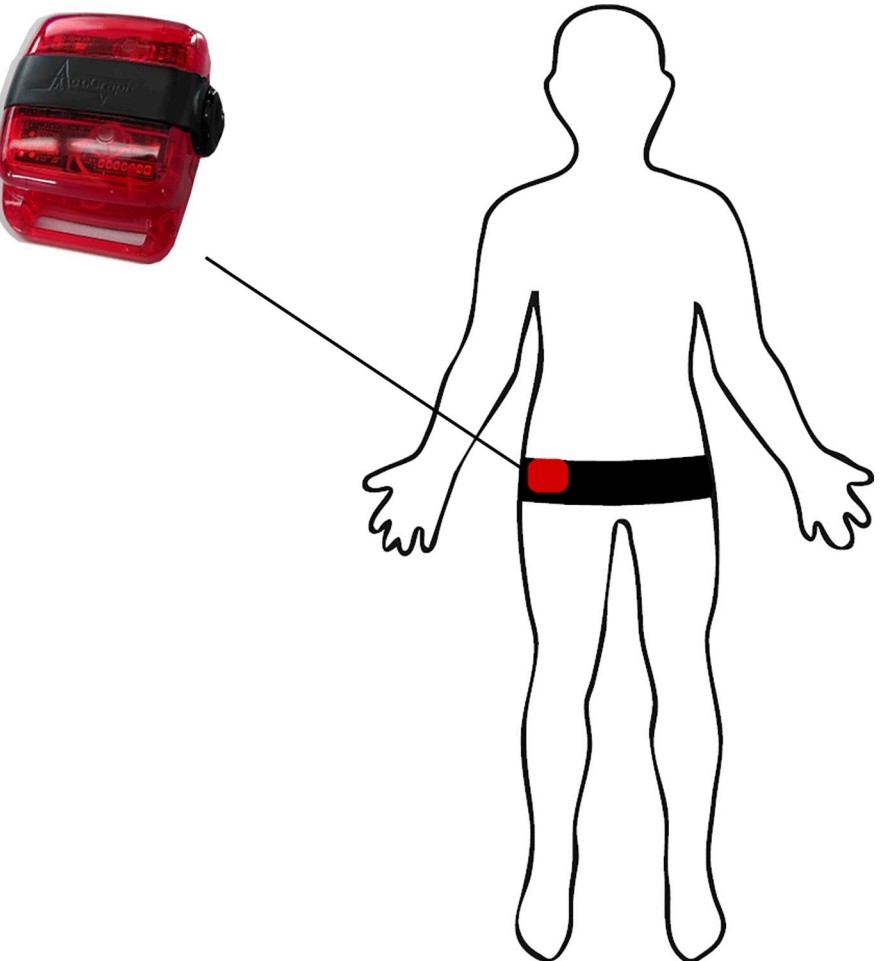

**Fig 1. Shows the accelerometer (the ActiGraph GT3X+) and how it was instructed to be worn.**

### Accelerometer data pre-processing

The accelerometer data was processed using the Python programming language, with the gt3x module used to read the raw accelerometer signal from file. From this we extracted the 10 second mean of the vector magnitude of body acceleration, calculated as the Euclidian Norm Minus One (ENMO) [20].This is calculated as shown in the equation below, where *accx, accy, accz* are the acceleration along the three orthogonal axes. We have used the ENMO because it automatically corrects for the contribution of gravity by subtracting one gravitational unit from the overall magnitude and has been shown to perform well in segmenting accelerometery according to activity intensity [20]. Following [20] we round negative values up to zero to prevent noise generating negative values.

$$ENMO = max\{\sqrt{(acc_x^2 + acc_y^2 + acc_z^2)} - 1, 0\}.$$

As the devices recorded movement at all times, regardless of whether the device was being worn or not, the first step in our data processing was to detect when the device was worn by the child in the morning and when it was last taken off at night. To do this, we segmented the recording into days starting and ending at 4am—following [11], calculated the activity to be

the mean of the acceleration in 10 second epochs—following [21], then took the first worn time to be the first epoch of non-transient activity, and the last worn time to be the last epoch of non-transient activity. We defined non-transient activity as activity recorded for at least 150 seconds out of every 1000 seconds. These parameters were determined empirically by applying a range of parameters to a sample of our recordings and selecting those that corresponded best with human selected wear times. We then detected any non-wear time during the day as any continuous period of zero activity longer than one hour [21]. As many of the children had worn the device for more than seven days, or had worn the device more sporadically, we then selected the seven continuous days that provided the maximum amount of wear time–the application of this algorithm to a sample recording is illustrated in S1 Fig. A recording day was only included in the analysis if it contained at least five hours of wear time; and the overall recording could only be included if it contained at least three days of suitable recordings [22].

## Training the HSMM

The HSMM model allowed us to segment and cluster our accelerometry traces according to the magnitude of acceleration. It can segment the trace into periods of similar acceleration, then cluster each segment into one of a number of hidden states. Hidden states are defined by their parameters (in this case parameters describing their acceleration distribution and duration distribution). The HSMM extends the better-known Hidden Markov Model (HMM) by explicitly modelling the duration of time spent in a state. This is modelled as a Poisson distribution and the observations (acceleration magnitudes) modelled by a Gaussian distribution. The trained model also contains a transition probability matrix, defining the likelihood of transitioning from one state to another. As the parameters for these distributions are learned in a Bayesian manner, the HSMM allows the hidden states to emerge solely from the data under study, unlike the traditional cut points method whose parameters must be derived from studies of external (and possibly unrepresentative) populations. We used the pyhsmm Python package [23] to train the HSMM, which implements a Hierarchical Dirichlet Process Hidden semi-Markov Model. The HSMM model requires us to specify a number of hyperparameters that influence the outcomes of the learning process as well as the computational resources required. We set the maximum state duration to 360 ten second epochs or 60 minutes. We set the maximum number of states to be six–this would allow a long duration and short duration state for each of the cut point categories (SED, LPA, MVPA). A condition for early stopping was specified to be when the Hamming distance between two consecutive iterations was less than 0.05, and a maximum of 20 iterations was used. The random seed was set to zero.

## Applying the cut-points approach

The cut points published by [7] are the most appropriate for our population in the currently available literature. They were measured using the same waist-worn triaxial ActiGraph GT3X + accelerometer as the children in this study, with the cut point of 40 suggested as a valid cut point for sedentary activity for typically developing children and for children with cerebral palsy who were considered ambulant. This study did not provide a cut point separating light physical activity (LPA) and moderate to vigorous physical activity (MVPA), so we have used cut points derived by [24,25] as the best available for this. These were also measured for a waist-worn ActiGraph GT3X+ accelerometer but were validated only with typically developing children. The cut points specified in the literature are in units of accelerometry counts rather than raw accelerations. To use the most appropriate cut points for our population, we processed the raw accelerometry into accelerometry counts, with an epoch of one second, using

the ActiLife software provided by ActiGraph. We then labelled each one second epoch using the following rules:

- Sedentary activity (SED) as less than eight counts per second (based on the estimate of 40 counts per 5 seconds [7]).

- Light physical activity (LPA) as more than eight counts per second [7] and less than 28 counts per second—based on an estimate of 420 counts per 15 seconds [24,25].

- Moderate to vigorous physical activity (MVPA) as more than 28 counts per second [24,25].

We then found bouts of continuous activity using a custom function written in Python based on the getBout.R function of the GGIR package [26]. This allowed 10% of an LPA bout or SED bout and 20% of an MVPA bout to be outside of the intensity range for that cut point (e.g. less than 28 counts per second for MVPA). By ignoring these interruptions, we are recognising that periods of activity may have short pauses.

## Results

The cut points approach and HSMM approach were applied to data from 279 children, with each child contributing between three and seven days of accelerometry recordings. Fig 2(A) shows the parameters of the six hidden states of the HSMM after training. The state duration is the λ of the Poisson distribution that models the duration of each activity state, the state amplitude is the mean of the Gaussian distribution that models the acceleration magnitude of each state. Here we can see that states with a high acceleration mean also have a shorter duration. This reflects the underlying feature that high intensity physical activity cannot be maintained for the same length of time as low intensity activity. We also see that the states can be clustered around three groups, high intensity-short duration (states 4 and 5), low intensity-long duration (states 2 and 3), and very low intensity (states 0 and 1)–the colour of each state reflects the group we have assigned it to. To facilitate comparison with the traditional physical activity intensity categories (SED, LPA, MVPA) we compare SED with states 0 and 1, LPA with states 2 and 3, and MVPA with states 4 and 5.

Fig 2(B) shows an example accelerometry trace for one day of activity along with an annotation indicating the HSMM states assigned to each segment as well as the classical cut points categories (SED, LPA, MVPA). Here we can see an example of non-wear time during the day (between 12:00 and 14:00) and we can see how the HSMM approach compares to the traditional cut points approach in application to a sample of accelerometer data.

Fig 2(C) shows the time spent in each of the HSMM states by the population of children. We can see that before 06:00 and after 23:00 most children are not wearing the device, but between these times the proportion wearing the device quickly rises to over 80%. There is a dip in the total children wearing the device around mid-day, likely to be attributable to nap time. The most active states (4 and 5) comprise about 20% of time spent throughout the day.

Fig 3(A) shows the extent to which the output of the traditional cut points approach overlaps with that of the HSMM approach, while Fig 3(B) also indicates the correlation between the outputs of each approach. We can see that SED and MVPA are strongly correlated with states 0–1 (rho = 0.67), and 4–5 (rho = 0.82), respectively. SED is negatively correlated with states 4–5 (rho = -0.28) and MVPA negatively correlated with states 0–1 (rho = -0.39). LPA does not show strong correlation with any state grouping.

To further assess the clinical utility of each approach, we compared the outputs to the four domains of the Paediatric Evaluation of Disability Computer Aided Test (PEDI-CAT) applied

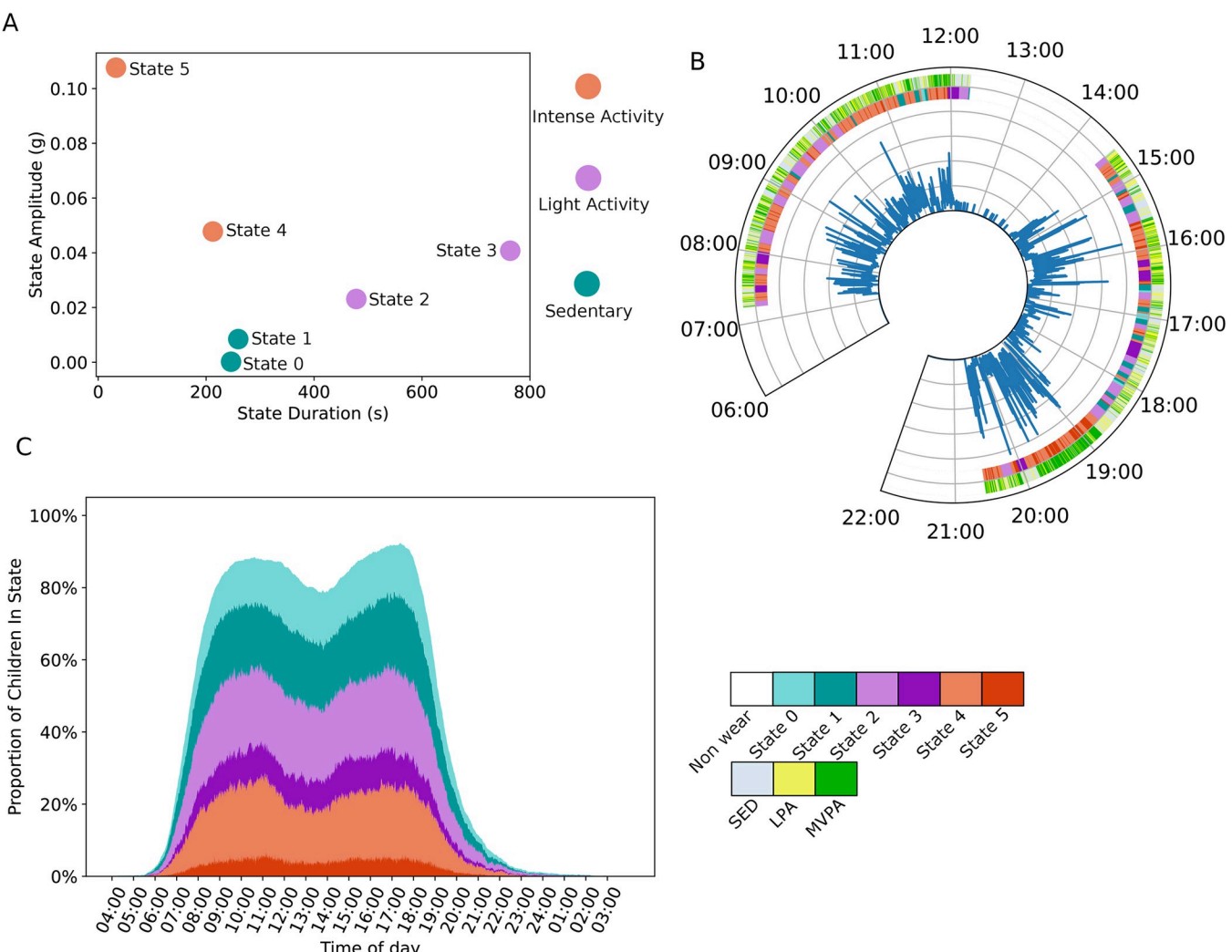

**Fig 2.** (A) Shows the parameters of each state in the HSMM. The colour corresponds to the group they have been placed in. (B) Shows a sample accelerometry trace from one day of recording (central trace) for one child and the corresponding classification according to the HSMM (inner circle) and traditional cut points approach (outer circle). The legend below indicates the states or cut point categories represented by each colour. (C) shows the proportion of children in each HSMM state throughout the day, with non-wear time shown as the area left white. The legend to the right indicates the colours corresponding to each state.

to the children from our sample. Fig 3(C) shows the linear regression of time spent doing MVPA (as a proportion of time wearing the device) and the score achieved by the child in each of the four PEDI-CAT domains. Fig 3(D) shows the same but for the time spent in the HSMM model states 4 and 5. Table 2 shows the coefficient of determination calculated from the above regression analyses, as well as a regression with age. The output of HSMM explains more variance in each of the four PEDI-CAT domains and age than the time spent in MVPA. Adding the time spent in all active HSMM states (2–5) did not improve the fit against age or any of the PEDI-CAT domains. Adding the LPA to the MVPA improved the fit slightly for PEDI Mobility ($R^2 = 0.42$), PEDI Activities ($R^2 = 0.27$), PEDI Responsibility ($R^2 = 0.14$) but states 4–5 of the HSMM remained a better fit for each.

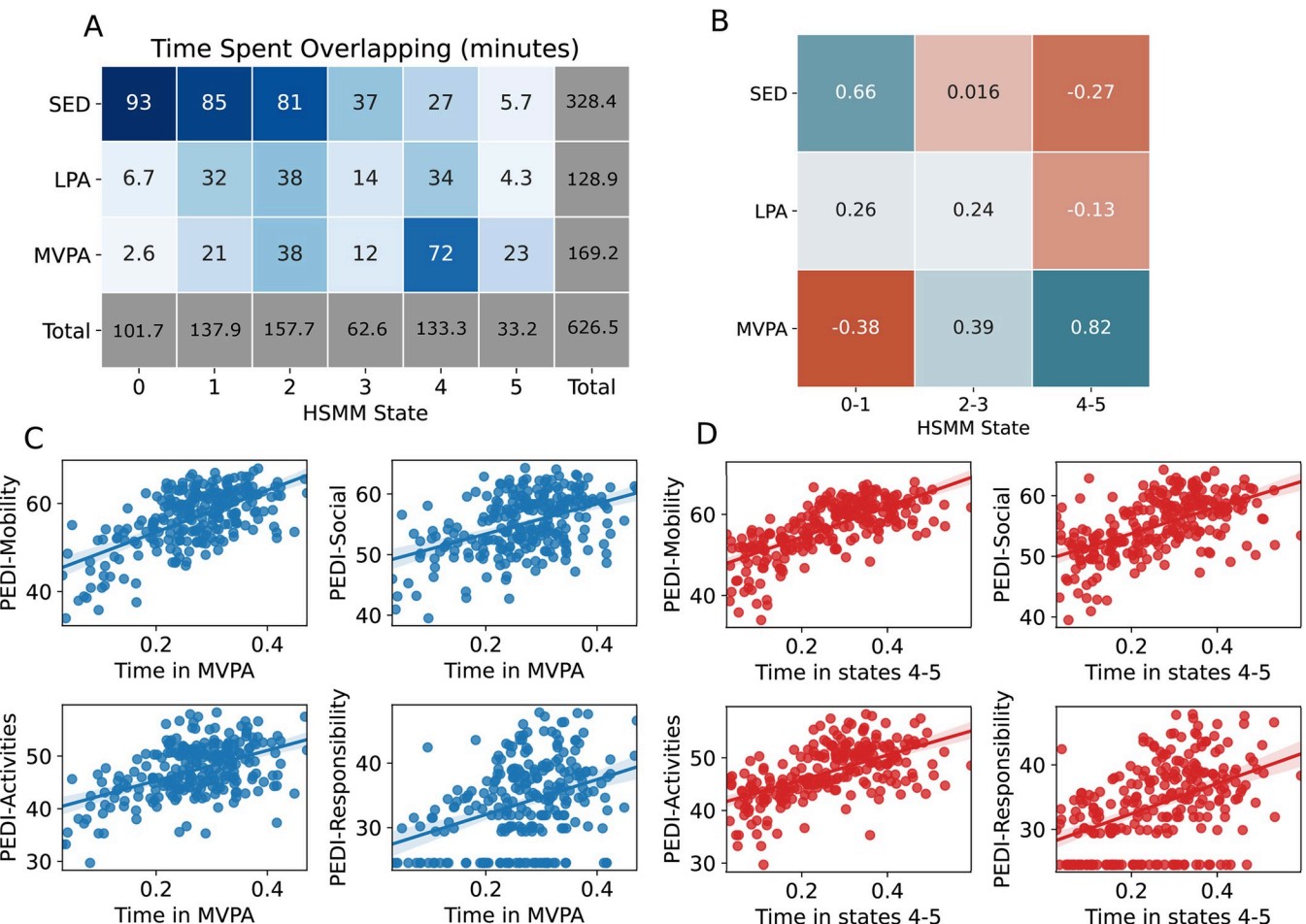

**Fig 3.** (A) Shows the mean time (in minutes) that each cut point derived category spends overlapping with each HSMM state. The states are sorted so that state 0 has the lowest mean acceleration and state 5 the highest. (B) Shows the correlation (Spearman's rho) between the time spent in each of the traditional cut points based physical activity intensity categories compared with the time spent in the grouped states of the HSMM. (C) Shows the linear regression of time spent doing MVPA against each of the PEDI-CAT domains. (D) Shows the linear regression of time spent in states 4 and 5 of the HSMM against each of the PEDI-CAT domains.

## Discussion

In the study reported, we have shown that hidden semi-Markov models can be used to segment and cluster multiday accelerometer data recorded in an ability-diverse population of

**Table 2. Shows the coefficient of determination ($R^2$) calculated on the linear regression of the time spent in HSMM states 4 and 5 or time spent in MVPA with age, and the four domains of the PEDI-CAT measure of ability.**

| Measure | $R^2$ for each method | |
|---|---|---|
| | Cut Points (MVPA) | HSMM (states 4–5) |
| Age | 0.1 | 0.15 |
| PEDI Mobility | 0.39 | 0.51 |
| PEDI Social Cognitive | 0.20 | 0.32 |
| PEDI Activities | 0.24 | 0.35 |
| PEDI Responsibility | 0.13 | 0.21 |

children aged 9 to 36 months. We have further shown that the resulting activity states are better predictors of the children's developmental capacity than the traditional analytical approach of using cut points.

Previous work [11,12] has shown that the HSMM can be used to segment and cluster wrist-worn accelerometer data recorded in teenagers or adults. We have built on this by demonstrating that the method can be used in young children (a rapidly developing and physically diverse population) and show that it produces an output that explains more of the variance in the children's PEDI-CAT domains, and so better captures their movement capacities than the traditional cut points approach.

The traditional cut points method used to quantify physical activity intensity from accelerometry relies on parameters derived from previous studies on external populations [24]. If these populations are not representative of those the method is applied to, there is a risk that the parameters may not be calibrated appropriately [3]. This can lead to the method failing to capture enough of the variance in physical activity behaviour present in the sample. For example, if the threshold for MVPA is calculated from a population of physically able adults and then applied to children with physical disabilities, the method would consider most of the population inactive most of the time. While they might rarely reach the physical activity intensities of able adults, there could well be important variations in this population that have not been detected. This can in part be overcome by calculating thresholds on the appropriate populations [7] however, this is expensive and is not feasible in all populations, e.g. in young children or in people with complex disabilities or health conditions [27,28]. Furthermore, even when appropriate thresholds are available, if one is studying a *diverse* population, or a rapidly evolving population, thresholds for different subpopulations or timepoints would be required, making it difficult to compare between groups and over time. This is a major challenge for longitudinal studies of young children or studies involving participants with a diverse range of physical activity behaviours [27]–often resulting in exclusion of these populations from research.

In contrast to the cut points approach[3], the HSMM learns the parameters for its hidden states from the data given to it [12,23], and the states can then be used to quantify and describe physical activity [11]. This approach has several potential advantages over the cut points approach. The parameters generated by HSMM can be easily interpreted–the mean of the Gaussian distribution (representing the observation distribution for the state) indicating the physical intensity of the state and the λ parameter of the Poisson distribution representing its duration. The time spent by a participant in each hidden state quantifies their physical activity participation. Our results further suggest the HSMM approach also has better clinical utility compared to the cut points, as the estimates of physical activity participation produced by the HSMM approach correlated more strongly with children's developmental capacity than the cut points estimates. Furthermore, it has the potential to make movement and physical activity research more inclusive for populations where calibration to energy expenditure is not possible–potentially reducing health inequalities over time.

## Limitations

In the present study we have used cut points to benchmark the HSMM approach. We have sought the most appropriate cut points for our population (young children with a diversity of abilities) and device type (ActiGraph GT3X+ worn around the waist). The LPA cut point taken from [7] partially meets these criteria, having been validated for pre-schoolers (both typically developing and with ambulatory cerebral palsy) wearing a GT3X+ around the waist. The MVPA cut point taken from [24] matches the device type and population age but has been

validated only for typically developing children. Our population however, contains children who are typically developing alongside those with a wide range of medical and developmental conditions (most of which neither of the cut points used have been validated for). This highlights the difficulty in applying the cut points approach to ability diverse populations–it would be prohibitively expensive to validate the cut points for each subgroup of our population–and further motivates the more flexible HSMM approach.

For regression with the PEDI-CAT domains and age we have taken the MVPA or HSMM states 4–5 rather than any physical activity (LPA + MVPA) or HSMM states 2–5. We choose this under the assumption that more vigorous forms of activity would relate more strongly to ability and age. This proved true for the regressions involving the HSMM output–in each case adding the time spent in less intense states (2–3) reduced the explained variance. For the cut points approach we found that for three of the PEDI-CAT domains (Mobility, Activities, and Responsibilities) the fit was improved slightly by adding the LPA, but not enough to make this combination a better predictor than states 4–5 of the HSMM. While the PEDI-CAT is not a direct and objective measure of physical activity participation, it has been shown to have discriminant validity with respect to the clinical assessment of a child's disability and provides a clinically valid assessment of a child's physical as well as cognitive and social behaviours [29,30]. The PEDI-CAT has also been shown to have concurrent validity with the Infant Toddler Activity Card Sort–a measure of activity participation in early childhood for children with developmental concerns [31]–and has been used to assess change in participation of activities in children as they receive physical therapy [32].

There are two key considerations to using the HSMM approach. First, it requires multiple iterations of its learning algorithm over the entire training dataset, and related high performance computing facilities and programming expertise. Second, the output cannot be directly associated with energy expenditure and thus its best suited for studies where direct estimation of energy expenditure is not a primary concern–such as in developmental studies with young children where movement and physical activity per se are the primary focus, or where additional measures are used to assess subsequent health and biological outcomes.

## Conclusion

The cut points approach continues to be an important tool for estimating energy expenditure from accelerometry. However, we have shown here that the HSMM approach can be used in an ability-diverse and rapidly developing population to provide a measure of physical activity participation. We hope this facilitates physical activity research in populations where cut-points are not currently available or populations for whom cut-point estimation is difficult.

## Supporting information

**S1 Fig. Illustrates the application of the algorithm used to identify wear time and to select 7 days of recording for subsequent analysis.** Red stars indicate the 7 days selected; the white line indicates wear time.
(TIF)

## Author Contributions

**Conceptualization:** Christopher B. Thornton, Niina Kolehmainen, Kianoush Nazarpour.

**Data curation:** Christopher B. Thornton, Niina Kolehmainen.

**Formal analysis:** Christopher B. Thornton.

**Funding acquisition:** Niina Kolehmainen.

**Investigation:** Christopher B. Thornton.

**Project administration:** Niina Kolehmainen.

**Resources:** Niina Kolehmainen.

**Software:** Christopher B. Thornton.

**Supervision:** Niina Kolehmainen, Kianoush Nazarpour.

**Visualization:** Christopher B. Thornton.

**Writing – original draft:** Christopher B. Thornton.

**Writing – review & editing:** Christopher B. Thornton, Niina Kolehmainen, Kianoush Nazarpour.

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
