## [Decision Letter · Decision Letter 0]

12 Dec 2022

PDIG-D-22-00297

Using unsupervised machine learning to quantify physical activity from accelerometry in a diverse and rapidly changing population

PLOS Digital Health

Dear Dr. Thornton,

Thank you for submitting your manuscript to PLOS Digital Health. After careful consideration, we feel that it has merit but does not fully meet PLOS Digital Health's publication criteria as it currently stands. Therefore, we invite you to submit a revised version of the manuscript that addresses the points raised during the review process.

Please submit your revised manuscript within 60 days Feb 10 2023 11:59PM. If you will need more time than this to complete your revisions, please reply to this message or contact the journal office at digitalhealth@plos.org. Please include the following items when submitting your revised manuscript:

We look forward to receiving your revised manuscript.

Kind regards,

Ryan S McGinnis, Ph.D.

Academic Editor

PLOS Digital Health

Journal Requirements:

a) Please clarify all sources of financial support for your study. List the grants, grant numbers, and organizations that funded your study, including funding received from your institution. Please note that suppliers of material support, including research materials, should be recognized in the Acknowledgements section rather than in the Financial Disclosure. 

b) State the initials, alongside each funding source, of each author to receive each grant. For example: "This work was supported by the National Institutes of Health (####### to AM; ###### to CJ) and the National Science Foundation (###### to AM)."

c) State what role the funders took in the study. If the funders had no role in your study, please state: “The funders had no role in study design, data collection and analysis, decision to publish, or preparation of the manuscript.”

2. We ask that a manuscript source file is provided at Revision. Please upload your manuscript file as a .doc, .docx, .rtf or .tex.

3. Please provide separate figure files in .tif or .eps format only and remove any figures embedded in your manuscript file. Please also ensure that all files are under our size limit of 10MB.

For more information about how to convert your figure files please see our guidelines: https://journals.plos.org/digitalhealth/s/figures

4. Please ensure that you refer to Figure 1 in your text as, if accepted, production will need this reference to link the reader to the figure.

Additional Editor Comments (if provided):

Reviewers' comments:

Reviewer's Responses to Questions

**Comments to the Author**

1. Does this manuscript meet PLOS Digital Health’s publication criteria? Is the manuscript technically sound, and do the data support the conclusions? The manuscript must describe methodologically and ethically rigorous research with conclusions that are appropriately drawn based on the data presented.

Reviewer #1: Yes

Reviewer #2: Partly

Reviewer #3: Partly

2. Has the statistical analysis been performed appropriately and rigorously?

Reviewer #1: Yes

Reviewer #2: Yes

Reviewer #3: No

3. Have the authors made all data underlying the findings in their manuscript fully available (please refer to the Data Availability Statement at the start of the manuscript PDF file)?

Reviewer #1: Yes

Reviewer #2: Yes

Reviewer #3: No

4. Is the manuscript presented in an intelligible fashion and written in standard English?

Reviewer #1: Yes

Reviewer #2: Yes

Reviewer #3: No

5. Review Comments to the Author

Reviewer #1: This paper investigates a Hidden Semi-Markov modelling approach for estimating activity levels in small children, going beyond traditional cut-point based approaches. The paper is relatively brief but is generally well written and presented with clear results and conclusions drawn. For improving the work I would suggest:

(1) The authors state participants were recruited from "the universal healthy child pathway (i.e. health visitors)". I think this is a very UK focused phrasing and I would suggest internationalising the terminology used to make the work more accessible. Equivalent terms in other countries are given at https://ihv.org.uk/our-work/international/health-visiting-across-the-world/. 

(2) In the ENMO equation, the opening bracket is in the wrong place, it should be after the max. Moreover, why is the max operation needed? If the device is stationary it should be recording 1g and so this will evaluate to 0 anyway. If the device is moving, the acceleration should be >1g. Is this just to compensate for noise when the device is stationary which might lead to brief negative values?

(3) More information needs to be given around the chosen cut point values, in the text around line 182. It is stated many times that the "the best available thresholds for the population" are used, but with no discussion around this. For example, I see that the device used in [7] was worn at the waist, as in your study, but only by going and reading [7]. This information should be included here. In the discussion there should be a section on the thresholds used, and in what ways they are appropriate and in what ways they represent a limitation (despite being the best available). 

(4) The caption for Fig. 2C needs improving - I struggled to see why the numbers didn't add up to 100% (until reaching this point in the text). Perhaps another colour for non-wear could be added.

Reviewer #2: The authors present a use case of the hidden Semi-Markov model (HSMM) with Bayesian inference to segment accelerometer data into physical behaviours in a paediatric population. They compare the activities segmented with HSMM with ones segmented using a cut point approach. They also compare the methods against domains of the PEDI-CAT, a report that evaluates disability in children.

Despite the use of HSMM not being novel, the authors provide evidence that the method can be used in a paediatric population with diverse motor abilities, evidence which is not currently available in the literature. 

Overall, the paper is clear and well written. My main comment is that more details could be provided. More detailed comments and suggestions can be found below.

Abstract

• How do you define “best available” in the sentence at lines 30-31? 

• Consider adding how the measures listed in lines 32-33 were measured (especially for daily activity as to avoid confusion with the accelerometer-measured one).

• Consider adding objective performance measures (e.g., correlation coefficient of unsupervised method VS one of cut-point approach).

Introduction

• While you reference previous works using Hidden Semi-Markov Models (references 11 and 12), from the introduction it is not clearly stated that other studies have used HSMM for activity segmentation purposes. You have acknowledged that HSMM have been previously used for this purpose in the Discussion, but I would suggest you also mention this in the introduction and clearly state how your study is different.

• Reference 11 and 13 are the same.

Material and methods

• In the paragraph from lines 109-114 and in Table 1, the sample size is not consistent (278 VS 279).

• It would be useful to provide more information on the PEDI-CAT assessment, e.g., format (is it a questionnaire?), what variables takes in consideration and what is the output. Alternatively, please provide a reference to consult if the reader requires further information.

• At line 154-156 it is mentioned that parameter were selected using a sample of recordings – are these recordings from the same dataset analysed in the rest of the paper or is it different data? 

• What is the rationale for modelling the durations with a Poisson distribution and the observations as a Gaussian distribution? Have you considered other distributions?

• Line 194: it was not clear to me what you meant with “interruption”.

• I suggest that you move Table 1 to the Results section.

Results

• In Figure 2B, I suggest using a different colour palette for the cut-point approaches as it is hard to distinguish the segments on the graph.

• Line 234: I suggest you don’t mention “agreement” as correlation cannot measure method agreement [1]. 

• In order to clinically validate the methods by regressing the time spent in activities with PEDI-CAT domains, there must be evidence that physical behaviours can predict PEDI-CAT domains. If this evidence is available, please provide references.

• Why hasn’t the regression analysis been conducted also for sedentary behaviour and LPA? Please explain the reasons in the text.

[1] Watson PF, Petrie A. Method agreement analysis: a review of correct methodology. Theriogenology. 2010 Jun;73(9):1167-79. doi: 10.1016/j.theriogenology.2010.01.003. PMID: 20138353.

Further comments

• Have you considered tuning the parameters of the observations/durations distributions? If not, what is the reasoning?

Reviewer #3: The study is interesting. Especially, the problem is well-stated. The manuscript, however, has several errors here and there. The manuscript should be prepared better. There is no conclusion and limitations of the study. The authors should read the manuscript carefully and fix all the grammatical and typos.

Abstract is written well. Still, the authors should mention what type of accelerometers was used? Where was it worn? Does the accelerometer provide raw acceleration data?

Line 65. Revise the sentence. Wearer’s is not commonly used, and sound weird. This should be also applied consistently to other sentences using similar terminology.

Line 91. I recommend that this sentence should be revised. I think you are not ‘wishing’ but demonstrating

Line 150. Perhaps a comma is needed after to do this, 

Accelerometer data pre-preprocessing: Can the data clarify how they have made these decision when processing the accelerometer data. Especially, proper references are needed. There is almost no references in this section, making it difficult to understand whether these methodological decisions are justifiable.

I was wondering if the authors could provide a figure to visualize how they cleaned the data and used HSMM for this problem.

Overall, I think the authors should provide references for many of their statements, especially in the methods. e.g Line 183

Line 193. I failed to understand what the authors mean by 10% of an LPA bour and 20% of an MVPA bour to be an interruption? 

Figure 3. Why the authors did not use states 1-3, when doing the regression? This also the case when looking at Table 2.

Again, the discussion needs improvement and proper reference. For example, Line 274-275, the authors say in contrast to cut points approaches, but there are no references in the whole paragraph

There is no conclusions. Also, limitation and strength should be also added.

6. PLOS authors have the option to publish the peer review history of their article (what does this mean?). If published, this will include your full peer review and any attached files.

**Do you want your identity to be public for this peer review?** For information about this choice, including consent withdrawal, please see our Privacy Policy.

Reviewer #1: No

Reviewer #2: No

Reviewer #3: No

---

## [Decision Letter · Decision Letter 1]

23 Feb 2023

Using unsupervised machine learning to quantify physical activity from accelerometry in a diverse and rapidly changing population

PDIG-D-22-00297R1

Dear Dr Thornton,

We are pleased to inform you that your manuscript 'Using unsupervised machine learning to quantify physical activity from accelerometry in a diverse and rapidly changing population' has been provisionally accepted for publication in PLOS Digital Health.

Best regards,

Ryan S McGinnis

Academic Editor

PLOS Digital Health

Reviewer Comments (if any, and for reference):

Reviewer's Responses to Questions

**Comments to the Author**

1. If the authors have adequately addressed your comments raised in a previous round of review and you feel that this manuscript is now acceptable for publication, you may indicate that here to bypass the “Comments to the Author” section, enter your conflict of interest statement in the “Confidential to Editor” section, and submit your "Accept" recommendation.

Reviewer #2: All comments have been addressed

Reviewer #3: All comments have been addressed

2. Does this manuscript meet PLOS Digital Health’s publication criteria? Is the manuscript technically sound, and do the data support the conclusions? The manuscript must describe methodologically and ethically rigorous research with conclusions that are appropriately drawn based on the data presented.

Reviewer #2: Yes

Reviewer #3: Yes

3. Has the statistical analysis been performed appropriately and rigorously?

Reviewer #2: Yes

Reviewer #3: Yes

4. Have the authors made all data underlying the findings in their manuscript fully available (please refer to the Data Availability Statement at the start of the manuscript PDF file)?

Reviewer #2: Yes

Reviewer #3: Yes

5. Is the manuscript presented in an intelligible fashion and written in standard English?

Reviewer #2: Yes

Reviewer #3: Yes

6. Review Comments to the Author

Reviewer #2: I thank the authors for submitting a revised version of the manuscript, in which the authors have addressed all my comments. The revisions have improved the article overall and strengthened its arguments.

Reviewer #3: The authors have addressed an interesting topic. Thanks for their interesting work.

7. PLOS authors have the option to publish the peer review history of their article (what does this mean?). If published, this will include your full peer review and any attached files.

**Do you want your identity to be public for this peer review?** For information about this choice, including consent withdrawal, please see our Privacy Policy.

Reviewer #2: No

Reviewer #3: No
